The antimicrobial activity of silver acetate against Acinetobacter baumannii in a Galleria mellonella infection model

http://orcid.org/0000-0001-9947-1205 Mannix-Fisher Eden
McLean Samantha samantha.mclean@ntu.ac.uk
School of Science and Technology, Nottingham Trent University , Nottingham , UK
Silva Pedro
Electronic publication date: 2021 Apr 22
Publication date: 2021
Volume: 9
Electronic Location ID: e11196
Received 2020 Dec 9; Accepted 2021 Mar 9
Copyright: © 2021 Mannix-Fisher and McLean
Copyright year: 2021
Copyright holder: Mannix-Fisher and McLean
License: This is an open access article distributed under the terms of the Creative Commons Attribution License, which permits unrestricted use, distribution, reproduction and adaptation in any medium and for any purpose provided that it is properly attributed. For attribution, the original author(s), title, publication source (PeerJ) and either DOI or URL of the article must be cited.
License URL: https://creativecommons.org/licenses/by/4.0/

Keywords: Acinetobacter baumannii, Galleria mellonella, Silver acetate, Infection model, Antimicrobial

Funding: Nottingham Trent University’s Vice Chancellors Award This work was supported by Nottingham Trent University’s Vice Chancellors Award. The funders had no role in study design, data collection and analysis, decision to publish, or preparation of the manuscript.

==============================
Background

The increasing prevalence of bacterial infections that are resistant to antibiotic treatment has caused the scientific and medical communities to look for alternate remedies aimed at prevention and treatment. In addition to researching novel antimicrobials, there has also been much interest in revisiting some of the earliest therapies used by man. One such antimicrobial is silver; its use stretches back to the ancient Greeks but interest in its medicinal properties has increased in recent years due to the rise in antibiotic resistance. Currently antimicrobial silver is found in everything from lunch boxes to medical device implants. Though much is claimed about the antimicrobial efficacy of silver salts the research in this area is mixed.

Methods

Herein we investigated the efficacy of silver acetate against a carbapenem resistant strain of Acinetobacter baumannii to determine the in vitro activity of this silver salt against a World Health Organisation designated category I critical pathogen. Furthermore, we use the Galleria mellonella larvae model to assess toxicity of the compound and its efficacy in treating infections in a live host.

Results

We found that silver acetate can be delivered safely to Galleria at medically relevant and antimicrobial levels without detriment to the larvae and that administration of silver acetate to an infection model significantly improved survival. This demonstrates the selective toxicity of silver acetate for bacterial pathogens but also highlights the need for administration of well-defined doses of the antimicrobial to provide an efficacious treatment.

Introduction

The use of silver as an antimicrobial has been described throughout recorded history (Alexander, 2009) and though its popularity waned with the development and widespread use of antibiotics from the 1940’s, the increasing global prevalence of antibiotic resistance amongst bacterial pathogens has reignited interest in this ancient remedy. Currently silver, in different forms, is used to reduce the incidence and severity of infection in wound treatment via the application of topical suspensions and dressings (Atiyeh et al., 2007; Politano et al., 2013) as well as being incorporated into indwelling medical devices such as vascular access grafts from companies including B. Braun Medical Ltd. (B. Braun Melsungen AG, https://www.bbraun.com/en/products/b/silver-graft.html) and Getinge AB. (Getinge AB, https://www.getinge.com/int/product-catalog/intergard-silver/). Despite its apparent popularity as an antimicrobial, the efficacy of silver treatments has met with mixed results in the research community (Chopra, 2007; Politano et al., 2013) and manufacturer efficacy claims are often difficult to verify due to a lack of access to raw data. Given these limitations on determination of antimicrobial activity, there is clear evidence that more independent data is needed to better understand the antibacterial activity of silver compounds currently in use for treatment of infection. Herein we investigated the efficacy of a commercially used silver salt, silver acetate, against a WHO priority 1: critical pathogen; carbapenem-resistant Acinetobacter baumannii.

Acinetobacter baumannii is a Gram-negative nosocomial pathogen. Its success as a nosocomial pathogen can be attributed to multiple factors including; the bacterium’s ability to adhere to and thrive on abiotic and biotic surfaces, its capacity to form biofilms and its multi-drug resistance arising from a variety of mechanisms (McQueary & Actis, 2011; Longo, Vuotto & Donelli, 2014). Patients in intensive care units are most vulnerable to A. baumannii infection with manifestations varying due to point of entry including urinary tract infections, bacteraemia, secondary meningitis, wound infections and most commonly, ventilator associated pneumonia (Coenye et al., 2008; McQueary & Actis, 2011). Many of these infections, such as those on catheters (Thallinger et al., 2014) and endotracheal tubing (Raad et al., 2011), are caused by the formation of biofilms increasing the difficulty for clinicians to clear the pathogen.

Acinetobacter baumannii, an ESKAPE pathogen (Enterococcus faecium, Staphylococcus aureus, Klebsiella pneumoniae, A. baumannii, Pseudomonas aeruginosa and Enterobacter species), is capable of multiple antibiotic resistances with many nosocomial strains isolated being multi-drug resistant and several strains identified as being pan-drug resistant (Kim et al., 2014). This led to the World Health Organisation assigning the carbapenem resistant strains of this species to the list of bacteria that pose the greatest threat to human health, specifically these strains are designated as Category I: critical, with the World Health Organisation encouraging prioritisation of research towards finding new means to combat infections caused by these strains (Tacconelli & Magrini, 2017). Despite the obvious threat posed to human health by this pathogen, A. baumannii is the least studied of the ESKAPE pathogens and little research has been undertaken to understand the efficacy of silver salts such as silver acetate against this organism. A. baumannii strains have numerous mechanisms of antibiotic resistance including intrinsic resistance, enzymes that alter the antibiotic thereby reducing or eliminating activity, efflux pumps, modification of drug targets and permeability defects (Lin & Lan, 2014).

Due to the rapid emergence of multi-drug resistant and pan-drug resistant strains of A. baumannii across the globe there is an increasing interest in development of antimicrobials including silver acetate as a potential means to combat these infections (World Health Organisation, 2015). Silver has a long history of antimicrobial activity due to the ionisation of silver in water and bodily fluids to produce the silver ion, Ag+ (Lansdown, 2006; Marx & Barillo, 2014). Silver ions have a widespread effect on bacterial cells, firstly by interacting with residues on the cell membrane including disulphides and phosphates to stimulate endocytosis of the ions. Once inside the cell, the ionic silver interacts with cell membrane enzymes to cause denaturation of the cell envelope and interacts with other vital enzymes that control respiration and replication (Marx & Barillo, 2014). Importantly, due to the multiple targets of silver within a bacterial cell the emergence of resistance is slow (Marx & Barillo, 2014).

Also important in developing efficacious infection treatments is the selective toxicity of the antimicrobial. Silver ions are thought to have low toxicity in the human body, often their accumulation after administration is transitory with minimal toxicity (Lansdown & Williams, 2004; Lansdown, 2006). Although silver ions have been found to accumulate in the organs and tissues of rats and humans at high doses (Drake & Hazelwood, 2005; Loeschner et al., 2011) and permanent accumulation of silver can occur in the cornea and skin after prolonged exposure, this is not thought to be life threatening (Lansdown & Williams, 2004; Lansdown, 2006). The uptake of silver into the body during silver treatment has not been investigated in depth, however clinical studies have shown increased absorption through partial-thickness burns (Wang et al., 1985; Boosalis et al., 1987; Coombs et al., 1992).

One means of better understanding host toxicity without the need for human or animal trials that has gained popularity in recent years utilises Galleria mellonella larvae models. Galleria mellonella, commonly known as the greater wax moth, are insects within the order lepidoptera. Their larvae have a rudimentary immune system that can be said to mimic the mammalian immune system (Hernandez et al., 2019). By comparison to mammalian models, larvae are cheap to source in large numbers and do not require ethical approval. They are easy to manipulate experimentally, can survive in temperatures suitable for investigation of human pathogens (25–37 °C) and require no specialised equipment for maintenance. To ensure standardisation amongst assays larvae should be obtained from reputable suppliers (e.g. Bio Systems Technology TruLarv™) that guarantee the larvae are not treated with antibiotics and their growth conditions upon receipt have been standardised, so that larvae arrive in the same life stage with similar approximate dimensions. This makes the use of Galleria larvae in simple animal models of toxicity, infection and antimicrobial treatment highly favourable (Tsai, Loh & Proft, 2016; Pereira et al., 2018; Hernandez et al., 2019).

The aims of this study were to determine antimicrobial efficacy of silver acetate against A. baumannii in vitro, to identify the concentration of silver acetate that causes toxicity in the G. mellonella model and to determine whether this antimicrobial can rescue the larvae from lethal A. baumannii infection.

Materials and Methods

Strains and culture methods for A. baumannii

Strains were obtained from the American Type Culture Collection (ATCC), USA and the National Collection of Type Cultures (NCTC), UK. Strains were stored at −80 °C as 20% glycerol stocks and were cultured on Mueller–Hinton agar and broth (Sigma Aldrich Ltd, UK) at 37 °C unless otherwise stated.

Silver acetate

Silver acetate was purchased from Sigma–Aldrich Ltd. Silver acetate stocks were prepared by dissolving in sterile distilled water and were stored protected from light at room temperature.

Minimum inhibitory and bactericidal concentration assays

Overnight cultures of A. baumannii were diluted to OD600 = 0.1 in sterile phosphate buffered saline (PBS). Wells of 96-well plates were filled with 100 µl Mueller–Hinton broth and silver acetate ((1,000 mg L−1) final) or meropenem ((10 mg L−1) final) was added to wells in column one as appropriate. Two-fold serial dilutions were performed across the silver acetate plates while meropenem was diluted in 1 mg mL−1 increments. Subsequently 10 µl diluted overnight culture was added to all test and positive control wells. Negative growth controls contained broth only. Plates were incubated statically at 37 °C for 18 h in aerobic conditions. The minimum inhibitory concentration (MIC) was determined as the lowest concentration of compound where no visible growth was observed. Minimum bactericidal concentrations (MBCs) were determined by plating out 10 µl spots of culture in triplicate onto Mueller–Hinton agar from every well where no growth was observed in the MIC assay. The MBC was determined as the lowest concentration of compound where growth could not be recovered.

Growth in the presence of silver acetate

Mueller–Hinton broth (0.9 ml) was added to the wells of a 24-well plate, with a further addition of 0.1 ml bacterial overnight culture diluted to OD600nm = 0.5. The plate was incubated at 37 °C with orbital shaking (4 mm) in a Biotek® citation 3 imaging reader until absorbance of wells reached an OD600nm of 0.3. Silver acetate was added to wells at final concentrations of 15.6, 7.80, 3.91, 1.95, 0.98 and 0 mg L−1. Growth was monitored every 20 min for 24 h and viability was measured every hour for the first 8 h.

Biofilm assay

Overnight cultures of A. baumannii were diluted to an OD600 = 0.5 in sterile PBS. Wells of 24-well plates were filled with one ml Mueller–Hinton broth and silver acetate was added to wells in column one to a final concentration of 7.8 mg L−1. Two-fold serial dilutions were performed across the plate. Subsequently 100 µl diluted overnight culture was added to all test wells and positive control wells. Negative controls contained broth only. Plates were incubated statically at 37 °C for 18 h in aerobic conditions. Thereafter broth was aspirated, and biofilms were washed three times in one ml sterile PBS. Wells were air dried and stained with 500 µl of 0.1 % (v/v) crystal violet for 1 h. Stain was then removed, and wells were washed with PBS. Stain bound to the biofilm was solubilised with 200 µl of ethanol. The OD540 of each well was measured using a BioTek® Cytation™ 3 cell imaging multi-mode reader.

Galleria mellonella

Galleria mellonella larvae were purchased from TruLarv™, Biosystems Technology, Exeter, UK and were used immediately upon arrival. Larvae selected for use were healthy as demonstrated by a melanisation score of four and moving freely without stimulation. For all incubations G. melonella were placed in vented petri dishes on Whatman™ filter paper. For each experiment, groups of 10 larvae were used per condition and all experiments were repeated a minimum of three times on different dates ensuring that each condition was tested with a minimum of 30 larvae from different batches of Galleria.

Inoculum testing

Cultures of A. baumannii NCTC 13302 were incubated overnight, then washed in PBS to remove residual media and diluted to the appropriate concentration. Healthy larvae were infected with bacterial cultures, equating to 1.7 × 102, 1.7 × 103, 1.7 × 104, 1.7 × 105, 1.7 × 106, or 1.7 × 107 CFU per larvae in a final volume of 10 µl by injection into the last left proleg. Control groups were injected with 10 µl PBS or were not injected. Larvae were incubated at 37 °C for four days and were monitored for melanisation score and survival every 24 h post-injection.

Toxicity assays

Healthy larvae were injected in the last left proleg with 10 µl water containing silver acetate at a final concentration of 1.25, 2.5, 5, 10, 20, 40 or 80 mg kg−1 animal weight. Control groups were injected with 10 µl water or were not injected. Larvae were incubated for four days at 37 °C with survival and melanisation scoring carried out every 24 h post-injection.

Treatment assays

Cultures of A. baumannii NCTC 13302 were incubated overnight, then washed in PBS and diluted to the appropriate concentration. Healthy larvae were injected with A. baumannii that equated to 1.7 × 105 or 1.7 × 106 CFU per larvae in the last left proleg. Following a 30 min incubation at 37 °C, a second injection was administered containing 0, 10 or 20 mg kg−1 animal weight of silver acetate into the last right proleg. Larvae were incubated for four days at 37 °C with survival and melanisation scoring carried out every 24 h post-injection.

Statistical analyses

Statistical analysis to determine the significance of difference in Galleria mellonella larval survival between different conditions used the Log rank (Mantel-cox) test. Analysis of biofilm formation and larval health scores used two-way ANOVA with Tukey’s multiple comparison.

Results

Silver acetate demonstrates antimicrobial activity against A. baumannii

To first demonstrate the antimicrobial activity of silver acetate five strains of A. baumannii were tested to determine the minimum inhibitory and bactericidal concentrations of this silver salt for each strain. The MIC of silver acetate was found to be 4.56 mg L−1 or lower for all strains tested, demonstrating the antibacterial activity of silver acetate against this species (Table 1).

To confirm whether the mechanism of antimicrobial activity of silver acetate was growth inhibition or killing in this species; MBC assays were performed. In all strains the MBC did not significantly differ from the MIC for silver acetate suggesting a bactericidal mechanism of action (Table 1).

Table 1 Minimum inhibitory and bactericidal concentrations for silver acetate against a range of A. baumannii strains demonstrates significant antimicrobial activity.

Standard MIC and MBC assays were performed against a range of A. baumannii strains grown in Mueller–Hinton Broth in a 96-well plate. Where the MBC appears to be higher than the MIC statistical analysis showed no significant difference (paired T-test P = 0.2839, N = 3 ± SD). Meropenem MIC’s determined for comparison to silver activity.

Bacterial/strain	Type/origin	Silver acetate	Meropenem	
MIC mg L−1 (SD)	MBC mg L−1 (SD)	MIC mg L−1
(SD)	
ATCC 17978	Clinical isolate	4.56 (1.59)	9.11 (3.19)	0.194 (0.0659)	
NCTC 12156	Type strain	3.91 (0)	7.81 (0)	0.556 (0.110)	
NCTC 13301	Type D carbapenemase reference strain OXA-23	3.91 (0)	6.64 (2.62)	>10 (0)	
NCTC 13302	Type D carbapenemase reference strain OXA-25	3.91 (0)	7.81 (0)	>10 (0)	
NCTC 13305	Type D carbapenemase reference strain OXA-58	4.56 (1.59)	13.03 (9.49)	5.47 (1.64))	

To evaluate the antimicrobial activity of silver acetate against A. baumannii to a greater extent the WHO priority 1: critical pathogen strain NCTC 13302 was chosen for further study.

The permanence of silver acetate toxicity against exponentially growing A. baumannii is concentration dependent

Frequently antibiotic intervention commences when the infection is established, and bacterial load is high with the pathogen actively growing. Considering this we sought to determine how exposure of exponentially growing bacteria to silver acetate at a range of concentrations impacted growth and viability.

Exponentially growing cultures of A. baumannii NCTC 13302 were established before the addition of varying concentrations of silver acetate that centered around the MIC for this strain. When cultures were exposed to concentrations of silver acetate below the recorded MIC (3.91 mg L−1) no significant decrease in growth was observed compared to the control (Figs. 1A and 1B). However, at concentrations above the MIC (7.8 and 15.6 mg L−1) loss of growth was rapid and permanent (Figs. 1D and 1E) Interestingly, exponentially growing cultures exposed to the MIC of silver acetate (3.91 mg L−1) showed a biphasic effect with inhibition of growth within the first 10 h after exposure but growth recovery over the following 14 h (Fig. 1C).

Figure 1 The growth inhibition of Acinetobacter baumannii exposed to silver acetate is concentration dependent.

Silver acetate was added to exponentially growing cultures at final concentrations of (A) 0.98, (B) 1.95, (C) 3.91, (D) 7.81 and (E) 15.60 mg L−1. Cultures were incubated at 37 °C with shaking and growth monitored via absorbance (OD600) every 20 min for 24 h. (F) Panel shows growth in the presence of all concentrations of silver acetate for comparison. N = 3, error bars are omitted for clarity, but standard deviations were all within the range 0.0006–0.1664.

To understand whether the silver acetate mechanism of action was bactericidal as suggested by the MIC and MBC assays (Table 1) viability assays were conducted on the cultures immediately prior to and at regular intervals after addition of silver acetate. Viability did not appear to be impacted when silver acetate was administered below the MIC but when administered at a concentration of double the MIC or higher there was a significant decrease in viability from two hours, with complete loss of viability at 6 h (Fig. 2). When exposed to silver acetate at the MIC, A. baumannii viability decreased 10-fold after two hours and 65-fold by 8 h, however these were modest decreases in viability that were recovered by 24 h (Fig. 2) suggesting that relatively few cells were killed in cultures exposed to this concentration of silver acetate and highlighting the need for appropriate dosing when using silver salts as antimicrobial agents.

Figure 2 The bactericidal activity of silver acetate against exponentially growing Acinetobacter baumannii is concentration and time dependent.

Batch cultures of A. baumannii were grown at 37 °C with shaking to early exponential phase when silver acetate was added at final concentrations of 0, 0.98, 1.95, 3.91, 7.81 or 15.60 mg L−1 (t = 0 h). Cultures were incubated for a further 24 h with viability determined at two-hour intervals to 8 h post-exposure (t = 2–8 h) and at 24 h. N = 3 ± SEM.

Biofilm formation in A. baumannii is inhibited by silver acetate

Biofilm formation is a major cause of infection and mortality. A. baumannii is known to readily form biofilms under numerous clinically relevant conditions, therefore we sought to determine whether silver acetate was effective in reducing biofilm formation.

Cultures of A. baumannii were incubated statically in a 24-well plate at 37 °C for 18 h in Mueller–Hinton broth containing concentrations of silver acetate ranging from 0 to 7.81 mg L−1. After incubation the amount of biofilm formation and number of viable cells were determined using crystal violet staining and viability assays respectively. At concentrations below the MIC, there was no significant difference in the amount of biofilm formed compared to the negative control. However, when bacteria were incubated with higher concentrations of silver acetate (3.91–7.81 mg L−), there was a significant reduction in biofilm formation (<0.0001, Fig. 3A). Enumeration of viable cells within the biofilms also showed no significant difference between the viability of cells exposed to silver acetate concentrations below the MIC compared to the negative control. At higher silver salt concentrations (3.91–7.81 mg L−) survival was reduced significantly (P < 0.005, Fig. 3B).

Figure 3 Biofilm production is significantly reduced in the presence of silver acetate.

A. baumannii cultures were incubated statically in Mueller-Hinton broth at 37 °C in 24-well plates in the presence of varying concentrations of silver acetate (0–7.81 mg L−1). (A) After incubation the wells were washed and stained with crystal violet to determine the amount of biofilm formed (OD540). Differences in biofilm formed between the lower (0–1.95 mg L−1) and higher (3.91–7.81 mg L−1) concentrations of silver acetate were highly significant (ANOVA and Tukeys multiple comparison test, ****P < 0.0001). (B) After incubation, biofilms were disaggregated and bacterial viability was determined. Significant differences in biofilm formation were observed between the lowest (0–0.488 mg L−1) and highest concentrations of silver acetate (3.9–7.8 mg L−1, ANOVA and Tukey’s multiple comparison, **P = 0.0005–0.003) N = 3 ± SEM.

Infection of G. mellonella with A. baumannii causes a lethal infection

Over the past decade G. mellonella larvae have become an established model for analysing the virulence and pathogenesis of human bacterial pathogens and in the testing of novel antimicrobial compounds. Initial testing in this study sought to determine the virulence of A. baumannii strain NCTC 13302 using a G. mellonella infection model. Here, the larvae were divided into groups and injected in the last left proleg with 10 µl of the appropriate number of bacteria (1.7 × 102–1.7 × 107 cells per larvae), PBS (vehicle control) or were not injected (no injection control). The larvae we incubated at 37 °C and monitored for changes in health every 24 h over the following four days. To ensure that batch variations did not bias the results, this experiment was repeated three times with different batches of larvae purchased on different dates. In total, 30 larvae per condition were tested.

Figure 4A shows survival of the larvae over four days, with the lowest concentration of bacteria injected causing no significant difference in larval death compared to the negative controls. The highest concentration of bacteria killed all larvae within 24–48 h and intermediate concentrations caused significantly different levels of killing over four days (P < 0.0001). The immune system of G. mellonella includes a cellular response called melanisation. This cellular response is used to trap microbes, but also makes it possible to track the immune response and health of the larvae via visible colour change (Wojda, 2017). Melanisation of the larvae was monitored as an indicator of health and supported the survival data with decreased health observed with increased bacterial load and significant variation in health across the higher concentrations of bacteria in comparison to the no infection controls (Fig. 4).

Figure 4 Infection of Galleria mellonella larvae with varying concentrations of Acinetobacter baumannii causes changes in health and lethality.

Groups of G. mellonella larvae were injected with 10 µl of A. baumannii containing between 1.7 × 102 and 1.7 × 107 cells per larvae, control groups were injected with PBS or were not injected. (A) Larval survival was monitored every 24 h for 96 h post-injection. The dotted line corresponds to 80% larval death (****P < 0.0001, N = 30 larvae per condition) (B) Melanisation was recorded for all larvae every 24 h for 96 h post-injection (*P = 0.0423, **P = 0.001–0.0063, ***P = 0.0006, ****P < 0.0001, N = 30 larvae per concentration, ±SEM). (C and D) Standard melanisation scoring.

A range of silver acetate concentrations commonly used in antibiotic therapy shows minimal toxicity to G. mellonella

For an antimicrobial to be appropriate for therapy it should display two key features: antimicrobial activity against the target pathogen(s) and minimal toxicity towards the host. We used a G. mellonella model to determine the toxicity of silver acetate towards the host over a variety of clinically relevant concentrations.

Galleria were divided into groups and were injected with silver acetate in concentrations ranging from 1.25 to 80 mg kg−1 of animal weight (approximately 0–24 µg per larvae). Negative control groups included larvae that were injected with water or were not injected. The experiment was repeated in temporally spaced triplicate to account for batch to batch variations in the larvae and collectively 30 larvae were tested per condition.

The data demonstrated that only the 80 mg kg−1 dosage of silver acetate caused persistent larval death of ~10% (Fig. 5A), although no significant difference between the survival of larvae at the different concentrations was observed (P = 0.0524). The melanisation scores showed that only larvae exposed to the highest concentration of silver acetate produced visible melanisation (Fig. 5B). A significant difference in melanisation developed between the 80 mg kg−1 injected larvae and no silver acetate controls from 72 h onwards.

Figure 5 A range of medically relevant silver acetate concentrations shows minimal toxicity to Galleria mellonella.

Groups of Galleria mellonella larvae were injected with 10 µl of silver acetate between 0 and 80 mg kg−1 animal weight. (A) Larval survival was monitored every 24 h for 96 h post-injection (N = 30 larvae per condition). (B) Melanisation was recorded for all larvae every 24 h for 96 h post-injection and assigned a standard melanisation score (***P = 0.0006, N = 30 larvae per condition, ±SEM).

Treatment of A. baumannii infection with silver acetate causes increased survival

To address whether silver acetate is an effective antimicrobial against infection in a G. mellonella infection model, larvae were injected with concentrations of bacteria that would cause significant death within four days without intervention (Fig. 3, 1.7 × 105 and 1.7 × 106 cells per larvae). Thirty minutes post-infection the larvae were administered with either 10 or 20 mg kg−1 silver acetate, both concentrations having demonstrated minimal toxicity towards Galleria (Fig. 5). As with previous experiments the Galleria were tested in groups of ten larvae. To ensure statistical significance of the results and to account for batch to batch variations with the larvae this experiment was repeated seven times with a total of 70 larvae tested per condition across all repeats.

Larvae injected with the ~105 bacterial cells were better able to survive than larvae injected with the ~106 bacterial cells in all conditions. Larvae treated with either concentration of silver acetate showed significant increase in survival (P < 0.05, Fig. 6). Larvae infected with the lower infectious dose of A. baumannii showed a 20% increase in survival after treatment with silver acetate (Fig. 6A). For larvae receiving the higher infectious dose, 10 mg kg−1 silver acetate treatment caused larval survival to increase by 31% and treatment with 20 mg kg−1 silver acetate increased larval survival by 27% (Fig. 6B).

Figure 6 Silver acetate treatment of Galleria mellonella larvae infected with Acinetobacter baumannii reduced lethality and improved overall health of the larvae.

Groups of Galleria mellonella larvae were injected with 10 µl of A. baumannii containing either 1.7 × 105 or 1.7 × 106 cells per larvae, 30 min post-infection groups of larvae were administered either 10 or 20 mg kg−1 silver acetate as treatment. N = 70 larvae per condition. (A and B) Survival and (C and D) melanisation were recorded every 24 h for 96 h post-injection. (A and B) *P < 0.05. (C and D) Error bars show SEM, *P = 0.0168–0.0235, **P = 0.0015–0.0089.

Correlating with the increased survival, improved larval health was also observed when silver acetate treatment was administered (Figs. 6C and 6D). As with the infection studies (Fig. 4) the melanisation score of larvae injected with both ~105 and ~106 bacterial cells showed a reduction in health from 48 h. However, after silver acetate treatment of Galleria infected with ~105 bacterial cells, differences in melanisation between the treated and non-treated larvae became significant after 72 h (P < 0.042). For Galleria infected with ~106 bacterial cells, significant differences appeared between the non-treated larvae and the larvae treated with both concentrations of silver acetate at 96 h (P < 0.009).

Discussion

Silver salts are incorporated into many commercially available, indwelling medical devices to provide antimicrobial activity during implantation for protection against infection during the time when the risk of infection is highest. Similarly, silver is also incorporated into wound dressings to provide antimicrobial activity against infected wounds (Leaper et al., 2012; National Institute for Helth & Care Excellence, 2020). The data presented herein sought to gain a better understanding of the antimicrobial efficacy and host toxicity of silver acetate to provide further insight into its potential value in infection control.

Silver acetate proved to be an effective antimicrobial in vitro for all A. baumannii strains tested with minimal inhibitory and bactericidal concentrations in the range of many clinically relevant antibiotics as assessed using standardised methods for determination of MIC and MBC values (The European Committee on Antimicrobial Susceptibility Testing, 2019). This included strains with limited antibiotic resistance and those with multiple resistances suggesting that existing antibiotic resistance mechanisms do not cause increased resistance to silver. Additionally, these values are broadly similar to reported MIC values for this species against silver nitrate (2.5 mg L−1, (Wan et al., 2016)) and silver ions (3.9 mg L−1, (Vaidya et al., 2017)). This is expected due to the antimicrobial mechanisms of silver acetate being caused by the silver ions in solution with the salt itself being largely inert in terms of antimicrobial activity. Silver ions are able to bind to the bases of DNA and RNA (Arakawa, Neault & Tajmir-Riahi, 2001), which is thought to be responsible for bacterial mutation and issues with replication, however this has yet to be proven in vivo. Silver ions cause numerous disruptions to proteins within the bacterial cell. Silver ions can bind to sulfhydryl groups on amino acids, disrupting protein function (Russell & Hugo, 1994), they are thought to disrupt iron-sulfur clusters (Xu & Imlay, 2012), thiol groups and sulfhydryl-liganded metals (Morones-Ramirez et al., 2013). There is also evidence of the role for silver in membrane disruption causing enhanced permeability (Morones-Ramirez et al., 2013; Vazquez-Muñoz et al., 2019). With this multitude of targets, it is perhaps unsurprising that antibiotic resistant strains are still susceptible to silver ions. Increased bactericidal activity has been reported via the production of silver nanoparticles. The size and shape of these silver nanoparticles play a key role in their bactericidal activity (Pal, Tak & Song, 2007).

The issue of resistance to silver compounds developing due to widespread use has been a topic of much debate. Whilst some research has suggested that the resistance to silver is slow to emerge and mild due to the requirement of the bacterium to develop resistances that nullify several of the above mentioned mechanisms of action (Marx & Barillo, 2014), others have described various mechanisms of silver resistance that is emerging across the globe. One such mechanism is the production of redox active metabolites, for example the production of pyocyanin by P. aeruginosa has been demonstrated to not only protect itself but also other species of bacteria in close proximity to the phenazine compound (Muller, 2018). This is a concern as P. aeruginosa is frequently isolated from polymicrobial infections. Another mechanism of silver resistance described in the literature is the increased expression or acquisition via horizontal gene transfer of the sil system (Hosny et al., 2019). This collection of genes primarily reduces intracellular accumulation of silver inside the cell by expressing periplasmic proteins that bind silver preventing further penetration into the cell and by expressing silver efflux pumps. This and the data provided herein highlights the importance of strict monitoring of silver use as an antimicrobial and administration of the appropriate dosages when used.

This study focused on the antimicrobial activity of silver acetate against the lesser studied ESKAPE pathogen; A. baumannii. The data herein demonstrates a similar trend in antimicrobial activity to this bacterial species as others have reported for more widely studied pathogens including S. aureus with MIC’s in the range of 0.08–32 mg L−1, P. aeruginosa with MIC’s of 0.04–8 mg L−1 and Escherichia coli with MIC’s of 0.5–2.5 mg L−1 (Peetsch et al., 2013; Zhang et al., 2015; Oates et al., 2018; Shah et al., 2020).

As well as the concentration at which antimicrobials can inhibit growth or kill bacterial pathogens another important property is the speed with which they are able to exert these effects. Herein we have demonstrated that exponentially growing cultures suffered a four-log reduction in viability at silver acetate concentrations similar to the minimal bactericidal concentration within 2 h exposure. Higher concentrations of the silver salt showed almost complete cell death by this time (Fig. 2) and 6 h post-exposure concentrations >7.81 mg L−1 caused a complete loss of viability. Other studies have also demonstrated both a time and concentration dependent killing of bacteria upon exposure to silver (Jaime-Acuña et al., 2016). Here, a silver-based nanocomposite was used to demonstrate the time and concentration dependent killing of E. coli.

Besides antimicrobial activity, a key factor in determination of whether silver acetate is suitable as an effective treatment for infection or prophylaxis is its toxicity to the host. Selective toxicity has been a cornerstone of antimicrobial therapy since Paul Ehrlich first proposed the concept by stating that the optimal agents would combine high parasitotropism with low organotropism (Witkop, 1999). G. mellonella is increasingly used as an ethically viable alternative to mammalian models for testing toxicity of compounds to the host (Dolan et al., 2016; Aneja et al., 2018; Cruz et al., 2018; Lazarini et al., 2018). One recent study compared the toxicity of exposure to eight different food preservatives in both G. mellonella and rat models, concluding that there was a strong correlation between the LD50 values of those preservatives in G. mellonella larvae and rats providing evidence of the suitability of this model for preliminary toxicity testing (Maguire, Duggan & Kavanagh, 2016).

In our study toxicity testing in the G. mellonella model revealed that only the highest concentration of silver acetate tested (80 mg kg−1 animal weight) caused significant death of the larvae. Therapeutic ranges of antimicrobials can be as low as 5 mg kg−1 daily to up to 85 mg kg−1 for urinary tract infections caused by indwelling catheters (National Institute for Health & Care Excellence, 2018). The dose depends on the administration method and the severity of the infection; however, most doses administered are less than 20 mg kg−1. As 10–20 mg kg−1 silver acetate showed no detrimental effects on the larvae here, it can be concluded that these are safe therapeutic doses in this model. Additionally, these low doses were able to significantly improve survival of the larvae after infection with a carbapenem-resistant strain of A. baumannii providing promising data for the clearance of drug-resistant bacterial pathogens.

The inoculum study revealed that ~106 bacterial cells per larvae were able to cause 80% larval death, which is an appropriate amount of mortality for subsequent treatment studies (Ignasiak & Maxwell, 2017). For comparison, a 10-fold lower infectious dose was also tested. The survival of larvae injected with ~105 bacteria cells increased by 20% at both concentrations of silver acetate used to treat the infection, while for the 10-fold higher infectious dose, survival increased by 27% and 31% for treatment with 20 and 10 mg kg−1 doses respectively. Future research could utilise this statistically significant improvement in survival upon administration of silver acetate in combination with antibiotic treatment to look for increased antimicrobial activity. Previous studies have demonstrated the potential synergy of silver ions with established antibiotics of the β-lactam, quinolone and aminoglycoside groups (Morones-Ramirez et al., 2013). More recently, silver nanoparticles have demonstrated synergy and additive effects when used in conjunction with antibiotics including kanamycin and chloramphenicol respectively (Vazquez-Muñoz et al., 2019). Here, membrane disruption was identified as a mechanism of action that improved antibiotic activity by allowing improved access to intracellular targets for these antibiotics. Further study of the adjuvant effects of silver acetate against drug resistant bacterial pathogens when administered alongside antibiotics currently on the market could provide a new route to antimicrobial treatment of these pathogens.

In conclusion the data presented here demonstrates the efficacy of silver acetate as an antimicrobial against carbapenem resistant A. baumannnii. We demonstrate that this silver salt is non-toxic to G. mellonella at concentrations able to cause significant antimicrobial activity and further that the administration of silver acetate can improve survival of infected larvae. Together this data suggests silver acetate is a suitable silver salt candidate for antimicrobial therapy when administered at an appropriate concentration.

Conclusions

Herein we sought to determine the antimicrobial efficacy of silver acetate against a carbapenem resistant strain of A. baumannii both in vitro and in an in vivo G. mellonella infection model. We found that silver acetate had bactericidal effects on the pathogen, was able to reduce biofilm formation and was able to significantly improve the survival of G. mellonella infected with otherwise lethal doses of A. baumannii. This data shows that silver acetate may be used as an effective antimicrobial at concentrations that are not damaging to the host and support the hypothesis that it can be used in efficacious antimicrobial therapy.

Supplemental Information

Supplemental Information 1 Antibiotic resistance profiles of A. baumannii strains using EUCAST clinical breakpoints (v11.0).

Disk diffusion assays were performed according to EUCAST standards. N = 3 ± SD, NP—clinical breakpoint not available from EUCAST. Shading: Green—sensitive, yellow—intermediate, red—resistant.

Click here for additional data file.

Supplemental Information 2 Raw figure data.

Click here for additional data file.

The authors would like to thank Hannah Southam for assistance in development of the Galleria mellonella models.

Additional Information and Declarations

Competing Interests

Author Contributions

Data Availability

The authors declare that they have no competing interests.

Eden Mannix-Fisher conceived and designed the experiments, performed the experiments, analyzed the data, prepared figures and/or tables, authored or reviewed drafts of the paper, and approved the final draft.

Samantha McLean conceived and designed the experiments, analyzed the data, prepared figures and/or tables, authored or reviewed drafts of the paper, and approved the final draft.

The following information was supplied regarding data availability:

Raw data is available in the Supplemental Files.

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
