# Peer review of "The antimicrobial activity of silver acetate against Acinetobacter baumannii in a Galleria mellonella infection model"

_PeerJ, doi:10.7717/peerj.11196_

## Round 0.1 · original submission · Major Revisions

Beyond some minor stylistic comments, Reviewer #1 has substantive concerns regarding the types of controls and number of replicates. Please address them appropriately. Reviewer #2 highlights the lack of novelty, but that is not a condition for publication in PeerJ, provided that the paper is otherwise a positive addition to the literature (e.g., it contains previously unknown facts/experimental support/etc., as your paper does). Reviewer #3 has provided their comments as a PDF, rather than as inline comments. Please ensure you do not miss their observations/requests.

From my own part, I would like to see the results with the requested controls but I do not insist on tests with "standard" antibiotics: I understand that you are mainly interested in using silver salts to circumvent resistance to traditional antibiotics, rather than to have "more powerful antibiotics". I would also like to see a comparison of your CH3COOAg results with other literature results on the use of silver salts and/or silver nanoparticles in this or other species.

Reviewer 1 ·

Basic reporting

1. Professional English has been used throughout the manuscript, some words are ambiguous and not clear. For example in line 42 (B. Braun Melsungen AG)(Getinge AB)
2. Literature references are not properly provided. For example, this reference is fine(Atiyeh et al., 2007)(Politano et al., 2013), but in lines 43 and 44 (Politano et al.,
44 2013)(Chopra, 2007), 2007 reference should be first. There should be uniformity in the literature references throughout the manuscript. Similarly this reference (McQueary & Actis, 2011)(Coenye et al., 2008). The references are not properly structured and have no sufficient field background.
3. The legends in the figures are not properly describing the figures. The fonts of the legends are different than the manuscript fonts.
4. The results are supporting, however there is weakness in the articles, as the control is just water and treatment is silveracetate, the silveracetate should be applied along with other positive controls for example commercial antibiotics against which bacteria were stilll not resistant.

Experimental design

1. The research is original within aims and scope of the journal.
2. Research question is well defined however along with silver acetate there should be one or two commercial antibiotics.
3. Lines 152-153, is not clear, 1.25 to 80 mg kg-1 animal weight. Controls included injection with the same volume of water and a non-injected group.
5. Investigation performed is not highly technical and ethically standard for example lines 157 and 158 lines are not clear Following the same general method as for inoculum testing 20 larvae were injected with 10 µl of 1.7 x 106 CFU per larvae of A. baumannii.
6. Treatment assays is confusing, the authors should describe with sufficient detail. Information should be properly added how many repeats, as there were 20 larvae, why not 50? why not 100?

Validity of the findings

1. Legends of the figures are not highly descriptive, as silver acetate is nowadays widely used, therefore the research not seem to be novel.
2. 166-169 These paragraph is ambiguous, five strains of A. baumannii were tested against varying concentrations of the metal salt to determine the minimum inhibitory and bactericidal concentrations for each strain. Silver acetate proved to be a potent antimicrobial with the compound having a minimal inhibitory concentration (MIC) of 4.56 ± 1.59 mg L-1 or less for 169 all strains tested (Table 1).
3. Conclusion is well stated that silver acetate may be effective to improve the survival of Galleria mellonella infected with otherwise lethal doses of A. baumannii.
4. Silver acetate is effective and what about the commercial antibiotics as positive controls, positive controls are missing.
5. Larvae from one source or from one company, why not from diverse areas or companies, might be the larvae are resistance to silver acetate as well as bacterial strains.
6. Why only bacterial strains of one species A. baumannii and why not authors have incorporated diverse type of bacterial strains?

Additional comments

1. You have used silver acetate to improve the survival of Galleria mellonella against lethal doses of A. baumannii, however silver acetate is only one nonantibiotics and there are no positive controls, there should be one or two commercial antibiotics.
2. You have used lethal doses of various A. baumanni, but not incorporated few of other bacterial species.
3. The larvae of the Galleria mellonella were purchased from one company or one area. What about the other larvae of the same species but might be genetically different.
4. These experiments should be repeated thrice, however only one experiment and no repeats.
5. In conclusion, your research is though novel however due to lack of proper replicates, diverse type of bacterial strains and diverse type of larvae, and absence of commercial antibiotics as positive controls, I suggest to resubmit again with proper answers of my questions.

Reviewer 2 ·

Basic reporting

The manuscript is well written in clear unambiguous language

Experimental design

More experiments is needed to ameliorate the manuscript for example investigate the effect of the tested salt on CRAB biofilm.

Validity of the findings

The study lacks of novelty

Additional comments

Here the authors described the efficacy of silver acetate against carbapenem resistant Acinetobacter baumannii using Galleria mellonella larvae model to assess the toxicity and the efficacy of this silver salt. Authors report the selective toxicity of silver acetate for bacterial pathogens. This is study lack of novelty. To me this work is quite insufficient in term of results, to be published as an original article.

---

## Round 0.2 · accepted · Accept

I am very satisfied with your responses to the reviewer's queries. I am glad to accept your paper for publication in PeerJ